# Plant Functional Traits on Tropical Ultramafic Habitats Affected by Fire and Mining: Insights for Reclamation

**Celestino Quintela-Sabarís** [1,*], **Michel-Pierre Faucon** [2]**, Rimi Repin** [3]**, John B. Sugau** [4]**, Reuben Nilus** [4]**, Guillaume Echevarria** [1,5] **and Sophie Leguédois** [1]

[1] INRAE, Université de Lorraine, LSE, F-54000 Nancy, France; Guillaume.Echevarria@univ-lorraine.fr (G.E.); sophie.leguedois@univ-lorraine.fr (S.L.)

[2] AGHYLE, UP 2018.C101, SFR Condorcet FR CNRS 3417, UniLaSalle, 60026 Beauvais, France; michel-pierre.faucon@unilasalle.fr

[3] Research and Education Division, Sabah Parks, Kota Kinabalu 88806, Sabah, Malaysia; sparks.researchedu@gmail.com

[4] Forest Research Centre, Sabah Forestry Department, Sandakan 90715, Sabah, Malaysia; john.sugau@sabah.gov.my (J.B.S.); reuben.nilus@sabah.gov.my (R.N.)

[5] Centre for Mined Land Rehabilitation, SMI, University of Queensland, St. Lucia QLD 4072, Australia

\* Correspondence: tino.quintela.sabaris@gmail.com

**Abstract:** Biodiversity-rich tropical ultramafic areas are currently being impacted by land clearing and particularly by mine activities. The reclamation of ultramafic degraded areas requires a knowledge of pioneer plant species. The objective of this study is to highlight the functional traits of plants that colonize ultramafic areas after disturbance by fire or mining activities. This information will allow trait-assisted selection of candidate species for reclamation. Fifteen plots were established on ultramafic soils in Sabah (Borneo, Malaysia) disturbed by recurrent fires (FIRE plots) or by soil excavation and quarrying (MINE plots). In each plot, soil samples were collected and plant cover as well as species abundances were estimated. Fifteen functional traits related to revegetation, nutrient improvement, or Ni phytomining were measured in sampled plants. Vegetation of both FIRE and MINE plots was dominated by perennials with lateral spreading capacity (mainly by rhizomes). Plant communities displayed a conservative growth strategy, which is an adaptation to low nutrient availability on ultramafic soils. Plant height was higher in FIRE than in MINE plots, whereas the number of stems per plant was higher in MINE plots. Perennial plants with lateral spreading capacity and a conservative growth strategy would be the first choice for the reclamation of ultramafic degraded areas. Additional notes for increasing nutrient cycling, managing competition, and implementing of Ni-phytomining are also provided.

**Keywords:** community weighted means; functional traits; soil reclamation; technosols; ultramafic

## 1. Introduction

Ultramafic soils are ecological or 'edaphic islands' due to their patchy distribution and contrasting soil conditions with respect to surrounding 'normal' soils [1,2]. Several extreme soil factors including macronutrient deficiency (N, P, K, Ca), Mg toxicity resulting in extremely low Ca:Mg molar ratio, and highly plant-available trace elements (Ni, Cr, Co) make ultramafic areas a stressful environment for plant establishment and growth [3,4]. The extreme edaphic conditions and isolated island-like distribution of ultramafic soils has led to the origin of numerous strict ultramafic endemic plant species, particularly in tropical regions, such as Cuba, New Caledonia, and Southeast Asia [5–7]. Ultramafic

areas in Sabah (North of Borneo, Malaysia) support a rich flora with more than 4500 species described to date and a very high proportion of strict endemics [8]. Besides its taxonomic and evolutionary interest [9], ultramafic flora is a remarkable biological resource for eco-technological applications, especially phytoremediation of contaminated soils [10].

Ultramafic areas have been extensively mined for the recovery of different metals such as Ni (Ni sulphide deposits and Ni laterites) and Cr (chromite) [11]. As the few high-grade Ni sulphide deposits have become depleted, mining for Ni has shifted focus to Ni laterite deposits in tropical areas including Australia, Cuba, New Caledonia, Brazil, and Indonesia [12]. In comparison to localized open pit mining of Ni sulphide, Ni laterite mining is highly destructive to ecosystems since it involves complete removal of vegetation and topsoil over a large area (strip mining) to access the Ni-rich saprolite below and some of the laterite [11,13]. Removal of the topsoil limits nutrient and water buffering capacity vital to the development of vegetation [11,14]. Logging and land clearing (mainly using fire) is another threat to ultramafic ecosystems, especially in Southeast Asia [7,15,16]. Logging and wildfires are less destructive than mining because they affect primarily the vegetation, leaving the soil more or less intact. However, after fires significant soil erosion and loss of carbon and some nutrients may occur. Tropical forest ecosystems may take more than a decade to fully recover after logging, whereas for mined areas it may take up to 250 years as the soil regenerates [17,18].

Pioneer plant communities on disturbed areas are derived from the local species pool and the effect of environmental filters (either stringent soil conditions or biotic interactions) [19]. It has been shown that experimental plant communities with different species composition subjected to similar environmental conditions experience a convergence in plant traits [20]. That is, successful plant species are those possessing the best traits that convey tolerance to the specific environmental stressors.

Plant functional traits are morpho-physio-phenological traits which affect fitness indirectly via their effects on growth, reproduction, and survival [19,21]. Trait-based ecology associated with trait data measured across many individuals and species can be used to predict emergent properties of communities and ecosystems. The functional trait approach allows for the characterization of plant responses to the environment [22] and their effects on ecosystem function and services such as nutrient availability or soil carbon storage [23,24].

Thus, the study of functional traits of plant communities that spontaneously colonize disturbed ultramafic areas may provide information for trait-assisted selection of candidate species for reclamation of tropical ultramafic degraded areas. An additional benefit of a functional approach is that the obtained information on traits can be transferred to other sites with similar environmental conditions, without the local/regional species pool limits that affect floristic approaches [25]. This approach has been previously applied to the revegetation of copper-cobalt mine areas [25], to the restoration of quarries [26], or to predict the colonization of post-mining sites during spontaneous revegetation [27]. Moreover, the combination of plant species with complementary traits has shown benefits for the phytoremediation of polluted soils and the restoration of mine tailings [28,29].

In order to examine the effects of disturbance severity on resulting vegetation type on tropical ultramafic areas, we studied the soil properties and plant communities of areas that experienced moderate disturbance (wildfire; FIRE) and severe disturbance (mining; MINE). Community weighted mean (CWM) represents the most probable value for a certain trait in a plant randomly sampled from a community [19]. Different studies based on CWMs have found changes in functional traits in relation to environmental factors, either on Mediterranean abandoned vineyards [21] or in tropical dry and tropical wet forests [30]. Here, CWMs were calculated to characterize functional response of communities according to type of habitat degradation and soil factors, and compared to the functional traits of general vegetation [31] and vegetation from non-disturbed ultramafic areas from Sabah [32].

Study goals included: (i) to examine how the type of disturbance (wildfire and mining) affects soil parameters and CWM of traits of pioneer plant communities on ultramafic soils, and (ii) to identify important traits for the trait-assisted selection of plant species for the reclamation of tropical degraded ultramafic areas.

## 2. Materials and Methods

### 2.1. Study Area

Ultramafic soils in Sabah (north of Borneo, Malaysia) cover around 3500 km$^2$ [33], with the more extensive ultramafic outcrops found around Mount Kinabalu, Morou Porou, Bidu-Bidu Hills, Meliau Range, Mount Tawai, and Mount Silam [8]. Our research sites were located on three ultramafic degraded areas southeast of Mount Kinabalu, including Garas-Lompoyou hill chain, Bukit Hampuan Forest Reserve, and Paliu area. These areas contain sites degraded by wildfire (FIRE) or quarrying (MINE) (Figure 1). Fires in Sabah are linked to El Niño Southern Oscillation (ENSO) drought events and affect mainly logged forest, which are more prone to fire than undisturbed forests [34]. Most of the FIRE sites were logged forests affected by severe fires which affected Sabah during the 1997/98 ENSO. MINE sites are the result of quarrying serpentinite bedrock for road base aggregate (abandoned in 1999) or rocky dumpsites (downhill sidecast) created during road construction in 2010. The four plots in Bukit Hampuan FR were in the vicinity of primary cloud forest on ultramafic substrate, whereas the plots in Garas-Lompoyou and Paliu were in a mosaic landscape composed by cultivated land and secondary vegetation in different degrees of succession (from fern-dominated areas to secondary forests).

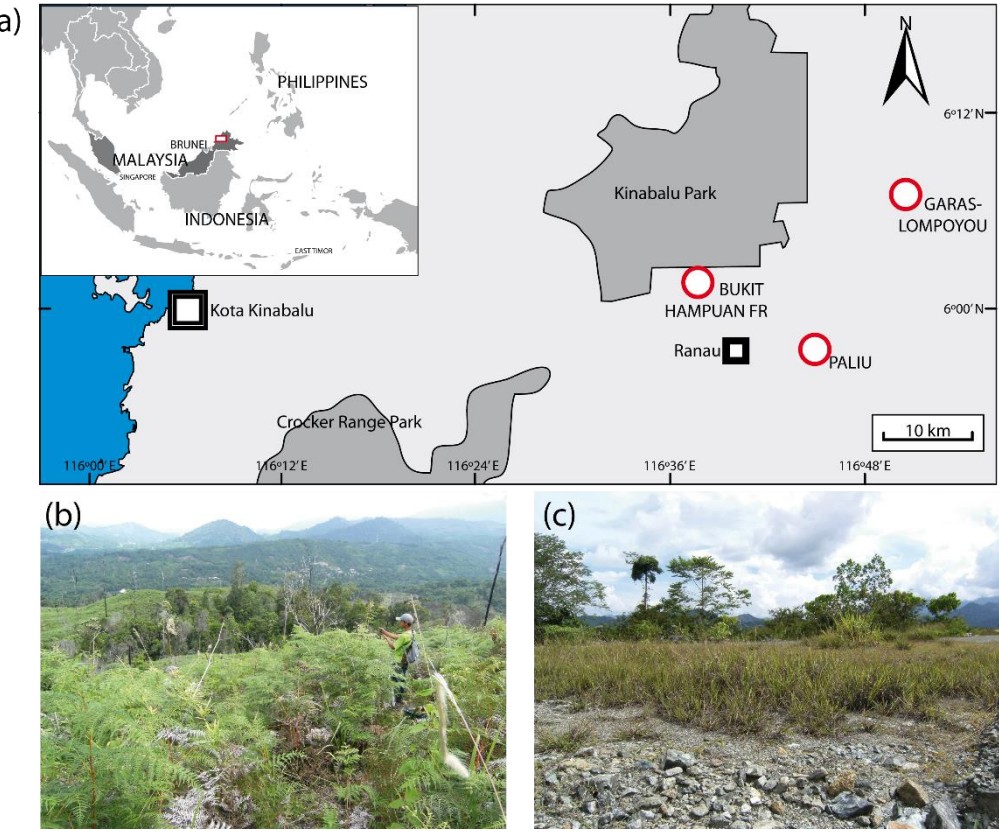

**Figure 1.** Overview of the studied area. (**a**) Position of sampling sites (marked as red circles). Main cities in the region are marked (with squares) for reference. Dark grey areas indicate Natural Parks. Position of the sampling area (red rectangle) in the context of South-East Asia is presented in the top-left insert. (**b**) General view of the vegetation in a FIRE plot. (**c**) General view of the vegetation in a MINE plot.

The climate of Sabah is tropical (Köppen climate Af). The mean annual temperature is 23 °C with low variation (less than 3 °C) throughout the year. Annual rainfall is around 2500 mm with relatively even rainfall throughout the year. Two notably less humid periods occur in February and August (see [35,36] and Malaysian meteorological department).

## 2.2. Plant and Soil Sampling

During July 2016, 15 circular non-permanent plots of 10 m radius were established on serpentinite-derived soils affected by soil excavation (MINE sites, eight plots) or by logging and wildfires (FIRE sites, seven plots). FIRE plots were in steep slopes (average 44%), whereas MINE plots were more flat (average slope 12%). Most of the plots had a south aspect. Plots in Paliu and Garas-Lompoyou area were in altitudes from 336 to 464 m asl, whereas the plots in Bukit Hampuan FR were around 1200 m asl. Soils in MINE sites were Spolic Technosols, whereas in FIRE sites soils were Cambic Leptosols [37,38]. In each plot, three radial transects separated by 120° were randomly established using a table of random numbers and a compass. In FIRE plots plant cover was estimated by line-intercept method, whereas in MINE plots vegetation cover was estimated by point-intercept method (one pin each 25 cm). This difference in methods was caused by the difference of vegetation height that made the use of the point-intercept method in FIRE plots not feasible. Both methods allowed the determination of percentage of cover for each species, as well as the percentage of bare soil.

In order to describe soil conditions, one representative soil sample (0–10 cm) was collected from the middle point of each transect (i.e., three soil samples per plot).

## 2.3. Soil Analyses

Fresh soil samples were sieved upon sampling and the >5 mm fraction was kept in plastic ziplock bags and stored for 8–10 weeks at 4 °C until analyses. The activity of four soil enzymes linked to the cycles of phosphorus (alkaline phosphatase), sulphur (arylsulphatase), carbon (β-glucosidase), and nitrogen (urease), and the hydrolysis of fluorescein diacetate (FDA, considered a proxy for the hydrolytic activity of the soil) were determined in fresh soil subsamples [39,40].

Soil enzyme activities were determined using colorimetric methods as indicated in [41]. The values of soil enzyme activities were expressed on an oven-dried soil basis.

Soil subsamples were air-dried and sieved at 2 mm. Water retention data were determined on a pressure plate apparatus for two water potentials (−10 and −15,800 kPa) [42]. Available water storage (AWS; g 100 g$^{-1}$), i.e., water disposable for plant growth was calculated by the following equation: AWS = (Wfc − Wwp), (1), where Wfc is the water content at field capacity (water potential: −10 kPa) (g 100 g$^{-1}$) and Wwp is the water content at permanent wilting point (water potential: −15,800 kPa) (g 100 g$^{-1}$).

Soil pH was measured in $H_2O$ using a 1:5 (v/v) ratio. Cation exchange capacity (CEC) was determined colorimetrically after treatment of the soil with a solution of cobaltihexamine trichloride 0.05 N [43]. The filtered soil:cobaltihexamine extracts were analyzed by means of Inductively Coupled Plasma-Atomic Emission Spectrometry (ICP-AES, Liberty II, Varian Inc, Australia) to determine the soil exchangeable concentrations of $Ca^{+2}$, $Mg^{+2}$, and $K^+$. Soil available phosphorus (Olsen-P) was extracted with a solution of $NaHCO_3$ and quantified by reaction with ascorbic acid [44]. Soil nickel availability was evaluated after extraction with DTPA-TEA at pH 7.3, 1:2 w/v, 2 h shaking) [45]. Soil subsamples were ground in a ceramic mortar. Total soil C and N was estimated by combustion in a CHNS analyzer (Vario Micro Cube, Elementar, Germany). Dry ground soil subsamples (0.5 g) were digested in 2 mL of concentrated $HNO_3$ and 6 mL of concentrated HCl on a hot plate at 105 °C. Final solutions were filtered (0.45 µm DigiFILTER, SCP science, Canada) and diluted to 50 mL with deionized water. Pseudo-total soil concentrations of Co, Cr, Mn, Ni, P, and S were estimated by ICP-AES (Liberty II, Varian).

## 2.4. Plant Analyses

Individuals from each species identified in each plot were sampled for the determination of 15 functional traits related to plant persistence, nutrient management, and tolerance to ultramafic soils. We chose these traits because of their interest in different aspects of reclamation of disturbed ultramafic areas: revegetation, limitation of erosion, implementation of phytomining, and soil nutrient improvement. Besides usual functional traits (such as specific leaf area—SLA), we included six elemental

concentration traits important to explain plant response to the particular conditions of ultramafic soils: Ni hyperaccumulation (Ni > 1000 µg g$^{-1}$); leaf tissue elemental concentrations of K, Ca, Mg, and Mn; and the Ca:Mg molar ratio of leaves (Table 1). Whole-plant traits (such as lateral spreading capacity or plant height) were assessed directly on the field in at least three plants per species. One plant per species was excavated to estimate rooting depth. Moreover, branches or shoots of 1–3 plants per species were collected, kept in sealed plastic bags, and transported to Monggis Substation (Kinabalu Park), where they were processed the day of the sampling. One healthy leaf per plant was selected and put below a glass layer and photographed with a digital camera. The camera was placed in a fixed support to guarantee its orthogonal position with respect to leaves. In some cases, the leaf was cut in several fragments to ensure the correct estimation of leaf area. All leaf photographs included a ruler to allow the estimation of leaf area. Leaf area was obtained from the digital photographs using ImageJ software [46]. After photographing, each leaf was cleaned with tap water, rinsed with deionized water, and put in a paper envelope. Leaf samples were kept in an oven (60 °C) for several weeks and dry leaf mass was obtained. Specific leaf area (SLA) was calculated as the ratio between leaf area and dry leaf mass, including petiole [47]. After dry weight was recorded, dry leaves were finely ground using a ball mill. Subsamples (0.5 g) of dry and ground tissue were digested at 95 °C in 2.5 mL of concentrated HNO$_3$ and 5 mL of H$_2$O$_2$ (30%). The final solutions were filtered (0.45 µm DigiFILTER) and diluted to 25 mL with deionized water. Leaf P, K, Ca, Mg, Mn, and Ni concentrations were measured by ICP-AES (Liberty II, Varian). Leaf C and N were quantified in dry ground leaves using a CHNS analyzer (Vario Micro Cube, Elementar, Germany).

*2.5. Data Analysis*

The soil dataset, including soil pH, soil water retention capacity (AWS, Wfc, Wwp), P-Olsen, total soil C and N, soil CEC, soil plant-available (exchangeable) Ca, K, and Mg, Ca:Mg molar ratio, DTPA-extractable Ni, pseudototal concentrations of P, S, Mn, Ni, Cr, and Co, and enzyme activities was analyzed using principal component analysis (PCA). PCA was based on a correlation matrix in order to account for differences in metrics among variables and no further standardization was applied. Differences in soil variables between type of degraded sites (FIRE vs. MINE) were further assessed by nested ANOVA analyses, with factors 'Type' and 'Site' nested within 'Type'. Plant species cover in each plot was used to compute Shannon's H diversity index. Differences in plant communities between FIRE and MINE sites were further assessed by non-metric multidimensional scaling (NMDS) using Bray–Curtis distances (hereafter referred as taxonomic NMDS), and constraining solution to only two dimensions. Shannon's index and NMDS were computed with functions *diversity* and *metaMDS* from the package *vegan* (ver 2.5-1) for R (ver. 3.4.4) [48].

A community weighted mean (CWM) was computed for each functional trait and for each plot applying the following formula: CWM = Σ ($p_i$ × $trait_i$), (2), where $p_i$ is the relative contribution of species *i* to the total plant cover of the community and $trait_i$ is the trait value of species *i* [21]. In the case of binary variables CWM indicates the frequency of the presence of a trait. In the case of ordinal variables, we kept the most common value in each plot. CWM has shown to be a reliable parameter that is not affected by differences in methods for the estimation of plant relative abundance or the trait values [49]. CWMs were computed using the functions *functcomp* from the package FD (ver 1.0-12) for R [50].

The CWM data on the 15 studied plots were used to perform a second NMDS analysis (hereafter referred as functional NMDS) using Gower distance with the function *metaMDS* from *vegan* package [48]. Solution was constrained to only two dimensions. Environmental factors (soil parameters, altitude, slope, aspect, time since disturbance) were fitted as vectors onto the taxonomic and functional NMDS ordinations using the function *envfit* from the package *vegan* [48]. The correlation of the environmental vectors with the NMDS ordination and the *p*-value of that correlation were estimated by 1000 permutations. Only the environmental factors with *p*-values < 0.05 were plotted onto the NMDS graphs.

Differences in taxonomic diversity (Shannon's Index) between MINE and FIRE sites were compared by one-way ANOVAs. Non-parametrical Mann–Whitney U tests were applied to compare CWM of binary and ordinal traits between MINE and FIRE plots, whereas one-way ANOVAs were used for the comparison of CWM of quantitative traits. When necessary, data were log-transformed to meet ANOVA assumptions. PCA, ANOVAs, and Mann–Whitney U tests were computed using SPSS (v. 15, SPSS Inc., Chicago, IL, USA).

**Table 1.** List of the plant functional traits assessed in the sampled species. For each trait we include the unit, the categories (for traits coded as binary or as ordinal variables), the associated ecological functions, and the interests for reclamation of degraded ultramafic habitats.

| Trait | Units | Categories/Domain | Associated Ecological Functions | Interest for Reclamation |
|---|---|---|---|---|
| Life cycle | Unitless | (0) annual (1) perennial | Response to disturbance and soil resources, competitive strength | Revegetation and/or limitation of erosion |
| Lateral spreading capacity | Unitless | (0) absence (1) presence | Competitive strength | |
| Depth of root system | In cm | (1) 0–10 (2) 10–30 (3) >30 | Response to disturbance and soil resources, competitive strength | |
| Plant height | In m | (1) 0–0.11 (2) 0.11–0.29 (3) 0.30–0.59 (4) 0.60–0.99 (5) 1–3 (6) >3 m | Response to disturbance and soil resources, competitive strength | |
| Density of stems | Number of stems in 1 $dm^2$ | (1) 1–10 (2) 10–30 (3) >30) | Competitive strength | |
| Specific leaf area (SLA) | $mm^2\,mg^{-1}$ | Positive decimal value | Response to soil resources, plant defense | Soil nutrient improvement |
| $N_2$ fixation | Unitless | (0) absence (1), presence | Response to soil resources, nutrient strategy | |
| Leaf N concentration (LNC) | $mg\,g^{-1}$ | Positive decimal value | Response to soil resources, influence in nutrient cycling | |
| Leaf P concentration (LPC) | $mg\,g^{-1}$ | Positive decimal value | Response to soil resources, influence in nutrient cycling | |
| Leaf concentrations of Ca, Mg, K and Mn | $mg\,g^{-1}$ | Positive decimal value | Nutrient strategy/response to ultramafic conditions | |
| Leaf Ca/Mg ratio | Unitless | Positive decimal value | Nutrient strategy/response to ultramafic conditions | |
| Ni hyperaccumulation | Unitless | (0) absence (1) presence | Response to ultramafic conditions | Phytomining |

## 3. Results

### 3.1. Soil Parameters

Observed values of soil variables are typical for ultramafic soils with pH around 7 (neutral); Ca:Mg molar ratio <1; low concentrations of important nutrients (e.g., P-Olsen concentrations lower than 3 mg $kg^{-1}$); high pseudototal concentrations of Ni, Cr, and Co (mean values around 2400, 3500, and 275 mg $kg^{-1}$, respectively) (Table 2). CEC is high to very high (mean values from 15 to 30 $cmol^+kg^{-1}$, Table 2), and Mg is the dominant cation on the exchange complex.

**Table 2.** Comparison of soil variables between MINE and FIRE sites. Second and third columns present average values (±SD) for each variable and type of disturbed site. The last column indicates the *p*-values for each comparison. *P*-values higher than 0.05 are considered non-significant.

| Soil Variable | Type of Disturbed Site | | *p*-Value |
|:---:|:---:|:---:|:---:|
| | **MINE** | **FIRE** | |
| pH H$_2$O | 7.89 (±0.59) | 6.64 (±0.52) | <0.001 |
| Soil water retention (g H$_2$O 100 g$^{-1}$ soil) | | | |
| Wfc | 26.2 (±9.4) | 46.6 (±13.4) | 0.008 |
| Wwp | 12.0 (±5.8) | 32.6 (±12.8) | 0.002 |
| AWS | 14.2 (±4.5) | 14.0 (±5.5) | 0.832 |
| C and N (mass %) | | | |
| Total C | 1.13 (±1.82) | 6.62 (±3.09) | <0.001 |
| Total N | 0.05 (±0.04) | 0.36 (±0.15) | <0.001 |
| C/N ratio | 19.1 (±13.8) | 18.1 (±3.8) | 0.573 |
| Pseudo-total concentrations of major and trace elements (mg kg$^{-1}$) | | | |
| P | 81.1 (±65.0) | 176 (±54) | 0.005 |
| S | 283 (±479) | 263 (±145) | 0.121 |
| Co | 125 (±62.2) | 434 (±172) | <0.001 |
| Cr | 1275 (±695) | 5826 (±2202) | <0.001 |
| Mn | 1516 (±610) | 4421 (±1082) | <0.001 |
| Ni | 1893 (±679) | 2941 (±1082) | 0.036 |
| DTPA-extractable Ni (mg kg$^{-1}$) | 18.9 (±17.4) | 155 (±62) | <0.001 |
| P-Olsen (mg kg$^{-1}$) | 0.59 (±0.47) | 2.83 (±1.85) | <0.001 |
| CEC and exchangeable cations (cmol+ kg$^{-1}$) | | | |
| CEC | 15.9 (±10.1) | 30.2 (±12.8) | 0.024 |
| Ca$^{2+}$ | 2.6 (±2.0) | 9.0 (±5.5) | 0.02 |
| Mg$^{2+}$ | 10.3 (±6.8) | 13.0 (±5.6) | 0.319 |
| K$^+$ | 0.1 (±0.1) | 0.3 (±0.2) | 0.004 |
| Ca:Mg | 0.4 (±0.3) | 0.9 (±0.8) | 0.052 |
| Soil microbial activities (µg product g$^{-1}$h$^{-1}$) | | | |
| Urease | 2.2 (±2.1) | 5.4 (±2.7) | 0.002 |
| Arylsulphatase | 5.3 (±7.0) | 80.2 (±30.7) | <0.001 |
| β-glucosidase | 48.7 (±13.7) | 82.6 (±20.4) | 0.001 |
| Alkaline phosphatase | 17 (±16) | 252 (±177) | <0.001 |
| FDA hydrolysis | 2.5 (±2.7) | 40.0 (±14.5) | <0.001 |

PCA analysis of soil data identified four principal components (PC) which explained 86% of variance in soil properties. The first PC explained most of the variance (58%). It was negatively correlated to soil pH and positively correlated to Ni-DTPA and most fertility factors: water retention parameters (Wfc and Wwp), FDA hydrolysis and enzyme activities, CEC and exchangeable K and Ca, Olsen-P, and total concentrations of C and N (Figure 2a). Pseudototal concentrations of Co, Cr, and Mn were positively correlated to PC1 and PC2 (11% of total variance). Exchangeable Mg had a little contribution to both PCs, whereas pseudototal S concentration was not correlated to any of these PCs (Figure 2a).

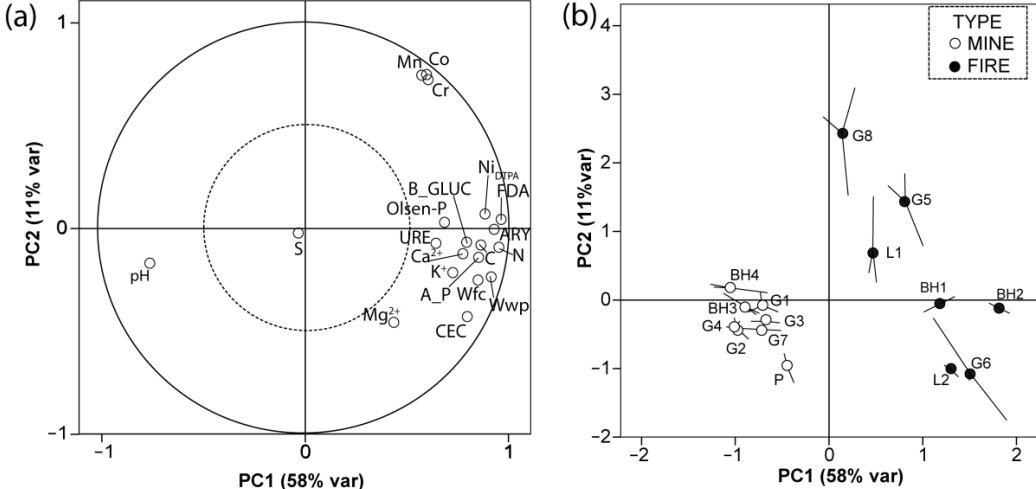

**Figure 2.** Principal component analysis (PCA) analysis of soil data. (**a**) Loadings of soil variables on principal components (PCs). Pseudototal concentrations of C, Co, Cr, Mn, N, and S are represented by their chemical symbol; A_P, Alkaline Phosphatase; ARY, Arylsulphatase; B_GLUC, β-glucosidase; CEC, cation exchange capacity; $Ca^{2+}$, $K^+$, $Mg^{2+}$, exchangeable concentrations of Ca, K, and Mg; FDA, hydrolysis of Fluorescein Diacetate; $Ni_{DPTA}$, available Ni; URE, Urease; Wfc, water at field capacity; Wwp: water at wilting point. (**b**) Scatterplot of soil samples (one point is the centroid of three samples per site, lines connect centroids with the position of each sample) on the two first PCs. Empty dots, MINE sites; black dots, FIRE sites. Codes indicate site: BH, Bukit Hampuan FR; G, Garas; L, Lompoyou; P, Paliu.

Projection of soil samples on first two PCs show a clear separation between FIRE and MINE plots. MINE plots are placed in a dense swarm on negative values for PC1 and ranging from −1 to 0.5 in PC2. FIRE samples had positive values in PC1 and were dispersed along PC2, from −1 to 3 (Figure 2b). Therefore, MINE plots had soils with higher pH and lower fertility than FIRE plots. However, the concentrations of trace elements were higher on FIRE than on MINE sites. Nested ANOVAs confirmed the inferences from PCA. For most of the analyzed variables, there were significant differences between site types (Table 2). It is remarkable that FIRE soils had better water retention properties (i.e., higher Wfc and Wwp) than MINE soils, but the available water storage (AWS) was similar between site types (around 14 g $H_2O$ per 100 g of soil, Table 2).

*3.2. Plant Communities*

A total of 42 plant species were sampled in the 15 studied plots (Table A1). Plant cover in MINE sites was lower than in FIRE sites (45% vs. 99%, *p*-value < 0.001). Number of sampled species per plot ranged from 2 to 11, whereas Shannon's Index ranged from 0.11 to 1.93. These values were similar in MINE and in FIRE plots (Table 3). The vegetation in MINE plots was dominated by different grass species (*Paspalum* spp. and others) with incidental presence of pioneer trees such as *Neonauclea gigantea* or *Ceuthostoma terminale* (present only in MINE plots from Bukit Hampuan) (Table A1). In FIRE plots the fern *Pteridium esculentum* was dominant, with minor presence of grasses (*Imperata cylindrica*, *Miscanthus floridulus*), pioneer trees (*Trema* sp., *Vitex* spp.), and gingers (family Zingiberaceae, present only in FIRE plots from Bukit Hampuan) (Table A1). The Ni-hyperaccumulator (*Phyllanthus rufuschaneyi*) was found growing on FIRE plots only. Several alien species were also found: *Mimosa pudica* (only in MINE plots), *Lantana camara* (only in FIRE plots), or *Chromolaena odorata* (in MINE and in FIRE plots). Results of taxonomic NMDS ordination showed a clear separation between the communities in MINE and in FIRE plots (Figure 3a). This separation along NMDS1 is mainly correlated to variation in soil properties (summarized by PC1, the first principal component of soil PCA, Figure 3a), whereas the vectors altitude, aspect, and time since disturbance were correlated (along NMDS2) to differences between Bukit Hampuan (BH) and the other plots within types of disturbed sites.

**Table 3.** Summary of plant cover and taxonomic diversity (number of species and Shannon's H index) in sampled plots in areas disturbed by fire (FIRE sites) or by quarrying/soil excavation (MINE sites). For each variable, the mean and the minimum-maximum (within brackets), are presented. *p*-values of one-way ANOVAs are presented in the last column.

| Variable | MINE Sites | FIRE Sites | *p*-Value |
|---|---|---|---|
| Plant cover (%) | 45.1 (21–84) | 99.5 (98–100) | <0.001 |
| N of Species | 5 (3–9) | 6 (2–11) | 0.322 |
| Shannon's H | 0.92 (0.18–1.93) | 1.05 (0.11–1.63) | 0.672 |

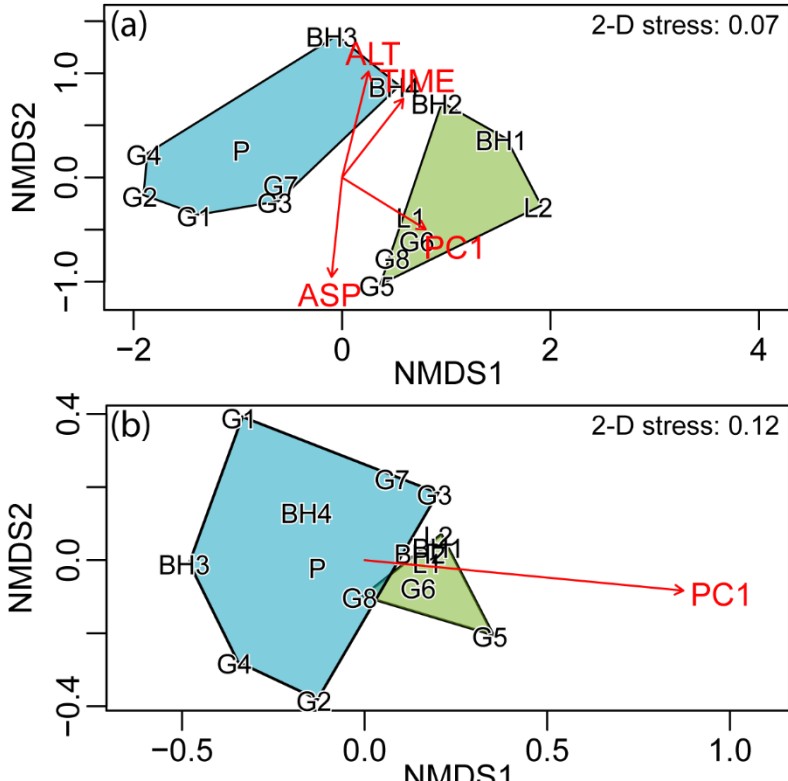

**Figure 3.** Non-metric multidimensional scaling (NMDS) of plant communities growing on degraded ultramafic areas in Sabah. (**a**) NMDS based on species composition. (**b**) NMDS based on community weighed means (CWMs) for 15 functional traits. Colored areas are convex hulls grouping MINE (blue polygons) and FIRE (green polygons) plots. The red arrows indicate environmental variables which are significantly correlated (i.e., *p*-value < 0.05) to the ordination. Abbreviations for environmental vectors are ALT, altitude above sea level; ASP, aspect; PC1, first principal component of soil PCA (see Figure 2); TIME, years from last disturbance. 2-D stress for each ordination is indicated in the graphs.

Regarding functional traits, similar values in CWM were obtained in both types of altered sites in 11 out of the 15 studied traits (Table 4). No variation was observed in life cycle (all sampled species were perennials). Depth of underground system was similar in the two types (0–30 cm). Plant height was less than 1 m in MINE plots as compared to 1–3 m in FIRE sites. Density of stems was higher in MINE sites, where communities were dominated by grasses/sedges. Plants with fixation of atmospheric $N_2$ were only present in MINE communities, whereas Ni hyperaccumulation was only present in FIRE plots, although low frequency of this trait made the differences between site types non-significant. Leaf traits (except Mn concentration) were similar between communities, whereas foliar Mn in MINE sites was two-fold higher than FIRE sites (Table 4). Compared to average values from the TRY database,

our studied plant communities had lower SLA and lower foliar concentrations of N, P, Ca, and Mn, whereas foliar K were in similar ranges and foliar Mg was much higher.

**Table 4.** CWM for 15 functional traits assessed in our study. For information on the type of variable, units and categories of each trait, please refer to Table 1. The second and the third columns present the CWM for each trait averaged for each type of site (MINE vs. FIRE), followed by the standard deviation (in the case of binary or quantitative traits) or by the maximum and minimum (in the case of ordinal traits). The fourth column indicate the *p*-values. We considered significant differences when $p < 0.05$. For comparative purposes, the last two columns present mean values of certain traits from the TRY database [31] and from an extensive study of ultramafic flora of Sabah [32].

| Trait | Type of Disturbed Site | | *p*-Value | Mean in TRY Database [1] | Mean in Sabah Ultramafic Flora [2] |
|---|---|---|---|---|---|
| | MINE | FIRE | | | |
| Life cycle (binary) | 1.0 (±0.0) | 1.0 (±0.0) | 1.0 | - | - |
| Lateral spreading capacity (binary) | 0.6 (±0.4) | 0.7 (±0.2) | 0.355 | - | - |
| Depth root system (ordinal) | 1.4 (1–2) | 1.9 (1–2) | 0.066 | - | - |
| Plant height (ordinal) | 3.4 (1–5) | 5.0 (5–5) | 0.015 | - | - |
| Density of stems (ordinal) | 2.1 (1–3) | 1.1 (1–2) | 0.006 | - | - |
| $N_2$ fixation (binary) | 0.1 (±0.1) | 0.0 (±0.0) | 0.016 | - | - |
| Ni_Hyperaccum (binary) | 0.0 (±0.0) | 0.01 (±0.02) | 0.285 | - | - |
| SLA ($mm^2\ mg^{-1}$) | 11.2 (±6.0) | 8.3 (±3.0) | 0.270 | 16.6 | - |
| Leaf N ($mg\ g^{-1}$) | 10.3 (±4.7) | 11.7 (±2.6) | 0.500 | 17.4 | - |
| Leaf P ($mg\ g^{-1}$) | 0.78 (±0.38) | 0.88 (±0.26) | 0.592 | 1.23 | 0.41 |
| Leaf K ($mg\ g^{-1}$) | 8.6 (±4.7) | 11.6 (±1.3) | 0.135 | 8.4 | 3.8 |
| Leaf Ca ($mg\ g^{-1}$) | 3.23 (±1.35) | 2.63 (±1.01) | 0.354 | 9.05 | 6.36 |
| Leaf Mg ($mg\ g^{-1}$) | 4.07 (±2.24) | 3.13 (±1.77) | 0.387 | 2.61 | 3.03 |
| Leaf Mn ($\mu g\ g^{-1}$) | 68.4 (±49.4) | 30.6 (±18.9) | 0.047 | 189 | 588 |
| Leaf Ca:Mg | 0.60 (±0.30) | 0.57 (±0.20) | 0.827 | - | - |

[1] TRY is an international database on plant functional traits. In 2011, it contained almost 3 million trait data entries for around 69,000 plant species [31]; [2] average values of flora from several undisturbed ultramafic sites of Sabah.

NMDS ordination of the studied plots on the basis of CWM showed that the FIRE and MINE plots partially overlap (Figure 3b). FIRE plots occupy less area (i.e., CWMs were similar) in the NMDS space, probably due to the dominance of *Pteridium esculentum* in those communities. Interestingly, only the soil conditions (summarized as the soil principal component PC1) are significantly correlated to this ordination. This fact indicates that the small differences in CWMs between communities are related to differences in soil properties.

## 4. Discussion

### 4.1. Soil Properties in Disturbed Ultramafic Habitats

In our study, we focused on tropical ultramafic areas disturbed by fire and excavation. Soil formation from serpentinite bedrock under tropical conditions leads to cambisols or to cambic leptosols with neutral to basic pH, very high CEC, high total and exchangeable Mg concentration, and high Ni availability [38,51]. In those soils, nutrients are scarce and mostly distributed in the upper soil horizons, where they are kept by an intense recycling of decaying organic matter [51]. The soils in MINE and FIRE sites of our study have the typical ultramafic properties described in previous lines. However, intense disturbance in MINE plots has eliminated the richer topsoil and resulted in a strong reduction of chemical fertility. Thus, soil carbon concentration is five-fold lower in MINE than in FIRE plots, the CEC of MINE plots is half of FIRE plots, the concentration of important nutrients is two-(P, K) to seven-(N) fold lower in MINE than in FIRE soils and the soil biological activity is extremely low in MINE plots. Soils derived from serpentinites are rich in 2:1 clays (smecites) [51] which have good water retention capacities. These clays could be responsible for the similar available water storage in MINE and FIRE soils, despite the important differences in soil organic matter content. Different studies have shown that ultramafic plants play a major role in the building up of Ni concentrations in the topsoil and its maintenance through biogeochemical recycling [51,52]. This phenomenon of plant

recycling could be responsible for the high pseudototal concentrations of trace elements (Ni, Cr, Co, Mn) and phytoavailable Ni in FIRE soils.

### 4.2. Functional Traits in Disturbed Ultramafic Habitats

Plants must deal with extreme conditions in ultramafic soils, which are even harsher in human-disturbed soils [3,4,11]. Our results on CWM for different functional traits allow the identification of main plant community adaptations to ultramafic stress. Despite the differences in soil conditions and taxonomic composition between MINE and FIRE sites, the CWM for most of the studied traits were similar in both types of disturbed areas. Moreover, the small differences we found were correlated to changes in soil properties (see 'functional NMDS'). In both types of disturbances, the plant communities had a conservative strategy (e.g., slow-growing species that conserve resources) [53]. As an example, CWM values of SLA are below the average SLA value in TRY database [31], and in the lower range of SLA values reported for tree species from tropical forest in Mount Kinabalu (SLA from 2.72 to 120.3 $mm^2$ $mg^{-1}$) [54] and for herbaceous plants growing on ultramafic substrates from Lesbos island (East Mediterranean) (SLA from around 10 to 45 $mm^2$ $mg^{-1}$) [55]. SLA has been correlated with relative growth rate and stress tolerance, as well as a protection against herbivores [56,57], thus we can conclude that our studied communities are characterized by low relative growth rates but high tolerance to stress. Community mean leaf concentrations of N, P, Ca, and Mn were slightly below the average values from TRY database and (for N and P) below the vegetation from Lesbos island [31,55]. In contrast, CWM values for leaf K and Mg are over the mean for TRY. High Mg CWM are understandable because Mg is the dominant element in exchange complex in serpentine-derived soils [38,51] and in our samples. However, leaf K concentrations (average 10.1 mg $g^{-1}$ K) are unusually high if we consider that the soils in the studied plots were deficient in this element. It is known that serpentinite-derived soils from Sabah are well drained and prone to drought [36,58]. Potassium plays a role in plant tolerance to drought [59], so these high levels of potassium could be an adaptation of pioneer ultramafic flora to cope with water limitation.

Regarding N and P, plant communities had intermediate leaf concentrations (11 and 0.83 mg $g^{-1}$ of N and P, respectively), which are lower than those observed in temperate ultramafic plants from Lesbos [55]. However, these concentrations are remarkable considering: (i) the extremely low soil concentrations of these nutrients, especially P (average N soil concentration 2.05 mg $g^{-1}$; average P-Olsen 1.7 mg $kg^{-1}$); (ii) the lower average leaf P concentration (0.41 mg $g^{-1}$) in plants from undisturbed ultramafic areas of Sabah [32]. Therefore, we may conclude that the plant communities in our studied plots are characterized by a high capacity of nutrient absorption and storage. Our findings in tropical degraded ultramafic areas are congruent with some of the functional characteristics previously described in ultramafic flora in temperate regions: slow growth rates, high investment in anti-herbivore defense, storage of nutrients, and efficient nutrient use [4,55]. This nutrient-conservative strategy is not restricted to ultramafic flora, but it is usually found in nutrient-limited ecosystems [53].

Soil Ca:Mg molar ratio has been identified as one of the important factors involved in the infertility of ultramafic soils, due to the antagonistic effect of high Mg concentrations over Ca uptake by plants [3,4]. In fact, the ability to maintain a leaf Ca:Mg molar ratio > 1 has been indicated as an important trait to explain adaptation to ultramafic soils in different ultramafic plants from temperate regions [60,61]. However, increased Mg requirements have been found in several temperate ultramafic plant ecotypes [3]. In contrast, leaf Ca:Mg molar ratios in our studied communities were around 0.5 (i.e., almost double of Mg moles in comparison with Ca moles in the leaves). Our results may be explained either by efficient tolerance mechanisms to excess Mg in plant tissues or higher Mg requirements in tropical ultramafic plants. However, more research is needed to clarify this topic.

The plant communities in MINE and FIRE plots differed in only four out of 15 studied traits: plant height, density of stems, $N_2$ fixation, and leaf Mn concentration. Plant height was higher in FIRE plots as a result of soils with more nutrient resources, but also as a response to high competition for light in sites with dense plant cover. The higher density of stems in MINE plots is just a consequence

of the dominance of grasses and sedges—with high number of culms—in these sites. $N_2$ fixation is an important characteristic, especially in nutrient-poor soils. Thus, it was unexpected that this trait appeared in very low frequency, and only in MINE sites. The difference in leaf manganese concentration is interesting as it cannot be explained by differences in soil Mn concentration: MINE sites had lower soil Mn and higher leaf Mn concentration than FIRE sites. Leaf Mn concentration has been proposed as a proxy to phosphorus-acquisition efficiency [62]: mechanisms for phosphorus mobilization at root level (release of carboxylates) provoke a significant increase in the absorbed Mn. However, to our knowledge, only one species from MINE plots (*Ceuthostoma terminale*, Casuarinaceae, with clusteroid roots) clearly has this strategy. The dominant species in FIRE plots (the fern, *Pteridium esculentum*) has very low leaf Mn concentrations (values from 5.5 to 11.5 mg kg$^{-1}$). Therefore, a more detailed study on the mechanisms of P-absorption would be needed to determine if this difference in Mn concentration in disturbed ultramafic communities is related to enhanced carboxylate release or whether it is just an artifact caused by the dominance of *P. esculentum*.

### 4.3. Implication in Revegetation of Ultramafic Degraded Areas

The analysis of functional composition of plant communities on degraded metal-rich soils may provide clues for the restoration of those areas, as well as trait-assisted selection of potential species for revegetation.

In our study we analyzed two types of tropical anthropized ultramafic habitats: serpentinite quarries or dumpsites and burnt areas. Soil conditions in these habitats mimic two possible ultramafic post-mining scenarios: (*a*) raw serpentine bedrock or saprolite exposed after soil excavation and (*b*) serpentine tailing amended with organic matter or covered with topsoil. On the basis of similar functional composition of tropical plant communities in FIRE and MINE habitats, we can conclude that suitable species for revegetation of scenarios *a* and *b* should possess the following attributes: perennial life cycle, lateral spreading capacity, rooting depth lower than 30 cm, and a nutrient conservative strategy (low SLA, high leaf K, and intermediate N and P concentrations). However, plants for scenario *a* should have between 10 and 30 stems per dm$^2$ and height between 0.3 and 0.59 m. In contrast, for scenario *b* the number of stems should be between 1 and 10 per dm$^2$ and the preferred height between 1 and 3 m. Finally, due to extremely low nutrient concentrations, revegetation in scenario *a* must include species with the ability of $N_2$ fixation [63,64].

Although some of these traits are present also in ultramafic plants from temperate areas, our results can be extrapolated only to tropical areas due to differences in serpentinite soils from tropical and temperate regions [51] and the specific conditions of tropical climate (e.g., high pluviometry, stable temperature, lack of marked dry season). As an example, high biomass producing temperate Ni-hyperaccumulators of the genus *Odontarrhena* spp. (formerly *Alyssum*) cultivated in a phytomining trial in Sulawesi (Indonesia) showed a low Ni concentration and a reduced biomass production [14].

Our results indicate that in case of extremely anthropized soil—scenario *a*—cultivation of selected plants in nursery and planting would be necessary to obtain a good plant cover (average plant cover in MINE plots is around 48%), whereas in scenario *b* spontaneous plant colonization may be an effective strategy for revegetation (almost 100% of plant cover in FIRE plots). However, spontaneous revegetation is feasible only if plant populations are locally present to act as seed sources [63] and if alien species are absent [65].

The use of species that fulfil the previous criteria in restoration of ultramafic degraded areas will reduce erosion (perennial plant cover, increased covered area due to lateral spreading capacity, limitation of surface runoff and sediment trapping due to dense number of stems) and nutrients will be conserved in the site. However, the obtained plant community will be poor in terms of functional diversity. Different studies have shown that the use of species with complementary functional traits have positive synergistic effects on the phytoremediation of multicontaminated soils and the reclamation of extremely degraded mine tailings [28,29]. With this aim, three additional elements should be explored to ameliorate restoration approaches in ultramafic habitats.

Nutrient cycling: plants with conservative strategies tend to have leaves with low SLA and to recover nutrients (i.e., nutrient resorption) during leaf senescence. Thus, produced litter is difficult to degrade and it contains reduced nutrient concentrations [23]. Measures to increase nutrient recycling, such as the introduction of $N_2$-fixing plants or inoculation of soil fauna (i.e., earthworms) should be explored [63,66].

Managing competition: the use of perennial plants with lateral spreading capacity can imply in some cases the selection of strong competitive species. Use of this species can create a favorable habitat for the colonization of other species (i.e., they can act as nurse species) or can lead to the creation of a dense monospecific plant cover that outcompetes any other species (e.g., degraded grasslands of *Imperata cylindrica* [34]). Depending on the desired outcomes, management would be needed to reduce competitive pressure of the dominant species. For instance, a sacrifice fallow crop with the exotic fast-growing *Acacia mangium* showed to be useful suppressing dominant *Imperata cylindrica* and creating microconditions for the germination and growth of native tree species [67].

Ni-hyperaccumulation: Ni-phytomining using hyperaccumulator plants has been proposed as being compatible with the restoration of mined areas [68]. Nickel would be recovered from the biomass of cultivated native hyperaccumulators and the obtained incomes would be used to cover (part of) restoration costs. The absence of Ni-hyperaccumulation from MINE plots indicates that phytomining would only be feasible in scenario *b*. A further improvement would consist of the selection of hyperaccumulators with other traits that fit conditions in degraded ultramafic areas (e.g., resprouting ability, resistance to drought, and full sunlight).

**Author Contributions:** Study design, C.Q.-S., M.-P.F., G.E. and S.L.; fieldwork organization and sampling, C.Q.-S., R.R., J.B.S. and R.N.; laboratory and formal analysis, C.Q.-S.; writing—original draft preparation, C.Q.-S.; writing—review and editing, all authors; supervision, C.Q.-S., G.E. and S.L.; funding acquisition, G.E. and S.L. All authors have read and agreed to the published version of the manuscript.

**Funding:** C.Q-S. contract and this research were funded by the French National Research Agency (ANR) through the national program "Investissements d'avenir" with the reference ANR-10-LABX-21-01/LABEX RESSOURCES21.

**Acknowledgments:** Sabah Biodiversity Centre provided C.Q-S. with the Access License ref no: JKM/MBS.1000-2/2 JLD.4(101). Sabah Parks and Sabah Forestry Department provided specific permits for the access to sampling sites. Romain Goudon, Lucas Charrois, and Stephane Colin are acknowledged for their contribution to the analyses of soils. Two anonymous reviewers provided valuable comments for the improvement of the manuscript.

**Conflicts of Interest:** The authors declare no conflict of interest. The funders had no role in the design of the study; in the collection, analyses, or interpretation of data; in the writing of the manuscript, or in the decision to publish the results.

## Appendix A

**Table A1.** List of plant species identified in 15 plots in degraded ultramafic areas from Sabah (Malaysia). Species codes are based in the first two letters of the genera and the specific epithet. When species identification was not possible, the codes correspond to the growing form: *F*, forb; *FE*, fern; *G*, grass; *T*, tree; *ZI*, ginger. Plant division is indicated in the third column when the plant family is not known. The last two columns indicate the number of plots where each species is present, and the relative cover (%) of that species averaged by the number of plots where it is present.

| CODE | Species | Division/Family | N Occurrences/Average Cover (%) | |
| --- | --- | --- | --- | --- |
| | | | FIRE plots | MINE plots |
| CETE | *Ceuthostoma terminale* | Casuarinaceae | 0/0 | 2/4 |
| CHOD | *Chromolaena odorata* | Asteraceae | 4/26 | 2/3 |
| CLSP | *Clausena* sp. | Rutaceae | 1/10 | 0/0 |

**Table A1.** *Cont.*

| CODE | Species | Division/Family | N Occurrences/Average Cover (%) | |
|---|---|---|---|---|
| | | | FIRE plots | MINE plots |
| COSP | *Colona* sp. | Malvaceae | 1/7 | 0/0 |
| COMSP | *Commersonia* sp. | Malvaceae | 1/1 | 1/3 |
| CY#01 | *Cyperus* sp. | Cyperaceae | 1/4 | 0/0 |
| DEFR | *Decaspermom fruticosum* | Myrtaceae | 1/1 | 0/0 |
| ETCO | *Etlingera coccinea* | Zingiberaceae | 1/4 | 0/0 |
| F#01 | - | Dicotyledon | 0/0 | 1/3 |
| FE#01 | - | Polypodiophyta | 0/0 | 2/2 |
| FE#02 | - | Polypodiophyta | 0/0 | 1/8 |
| FISP | *Fimbristylis* sp. | Cyperaceae | 0/0 | 4/35 |
| G#01 | - | Poaceae | 0/0 | 1/24 |
| G#02 | - | Poaceae | 0/0 | 3/24 |
| G#03 | - | Poaceae | 0/0 | 1/1 |
| G#04 | - | Poaceae | 0/0 | 1/26 |
| G#05 | - | Poaceae | 0/0 | 1/3 |
| G#06 | - | Poaceae | 0/0 | 1/3 |
| IMCY | *Imperata cylindrica* | Poaceae | 4/5 | 1/8 |
| LACA | *Lantana camara* | Verbenaceae | 1/10 | 0/0 |
| LYSP | *Lygodium* sp. | Lygodiacaeae | 3/4 | 0/0 |
| MA#01 | *Macaranga* sp.1 | Euphorbiaceae | 1/1 | 0/0 |
| MA#02 | *Macaranga* sp.2 | Euphorbiaceae | 1/4 | 0/0 |
| MA#03 | *Macaranga* sp.3 | Euphorbiaceae | 1/2 | 0/0 |
| ME#01 | *Melastoma* sp. | Melastomataceae | 1/7 | 0/0 |
| ME#02 | *Medinilla* sp. | Melastomataceae | 0/0 | 1/1 |
| MIFL | *Miscanthus floridulus* | Poaceae | 3/9 | 2/5 |
| MIPU | *Mimosa pudica* | Fabaceae | 0/0 | 3/5 |
| NASP | *Nauclea* sp. | Rubiaceae | 1/11 | 0/0 |
| NEGI | *Neonauclea gigantea* | Rubiaceae | 1/2 | 2/2 |
| PASP1 | *Paspalum* sp1. | Poaceae | 0/0 | 2/4 |
| PASP2 | *Paspalum* sp2. | Poaceae | 0/0 | 6/3 |
| PHRU | *Phyllanthus rufuschaneyi* | Phyllanthaceae | 1/5 | 0/0 |
| PTES | *Pteridium esculentum* | Dennstaedtiaceae | 7/63 | 0/0 |
| RU#01 | *Rubus* sp. | Rosaceae | 0/0 | 1/1 |
| T#01 | - | Dicotyledon | 1/2 | 0/0 |
| T#02 | - | Dicotyledon | 1/1 | 0/0 |
| T#03 | - | Dicotyledon | 1/3 | 0/0 |
| TRSP | *Trema* sp. | Cannabaceae | 2/3 | 0/0 |
| VIPI | *Vitex pinnata* | Lamiaceae | 1/5 | 0/0 |
| VISP | *Vitex* sp. | Lamiaceae | 1/5 | 0/0 |
| ZI#01 | - | Zingiberaceae | 1/4 | 0/0 |

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
