# Peer review of "Plant Functional Traits on Tropical Ultramafic Habitats Affected by Fire and Mining: Insights for Reclamation"

_diversity, doi:10.3390/d12060248_

Round 1

Reviewer 1 Report

The manuscript entitled " Plant Functional Traits on Tropical Ultramafic Habitats Affected by Fire and Mining: Insights for Reclamation" describes a study that investigates differences in soil attributes and species diversity and functional traits in serpentine habitats impacted by different types of disturbance. The study is experimentally well-designed, while the statistical analysis and the replication of the plots is adequate. In general, the manuscript is well-written and easy to follow. Although there is no major weakness in this study, the major comments/suggestions to the authors are related to a better description of the fire history of the burnt sites and of a more detailed description of the surrounding vegetation of the studied sites (serving as the seed source for the colonization of the plots after the disturbance).

Comments/Suggestions

Line 26: Please rephrase to “conservative growth strategy”

Line 35: Please rephrase to “highly plant-available”

Line 42: Please add reference(s)

Line 49: Rephrase to “Removal of the topsoil limits”, considering that if the capacity was completely removed there would be no vegetation at all after the disturbance.

Line 50: I would suggest replacing the adverb “healthy” with a more literal alternative.

Line 53: Please rewrite for better clarity.

Line 54: It could be rephrased as “Tropical forest ecosystems”.

Line 82: It should be either “Community Weighted Means represent the most probable values” or “Community Weighted Mean represents the most probable value”.

Line 84: Rephrase to “traits along ecological successions”. Also, replace “on” with “in”.

Line 85: In lines 81-82 you report that CMW means “Community Weighted Means”. If it doesn’t change, you don’t have to write CWMs to declare the plural.

Line 100: Replace “wildfire” with “wildfires”.

Lines 98-103: Did all sites have similar surrounding plant communities. The surrounding vegetation plays a tremendous role as source for the new communities colonizing the disturbed plots. Please provide some more information related to that.

Lines 100-101: The “frequency of fire” (how many times a site has been burnt in the past) and the “time since the last fire” are two very important parameters affecting the vegetation and thus the functional diversity of plant communities. The authors should provide some more info related to fire history of their plots.

Line 113: Rephrase to “of 10 m radius”.

Line 128: Rephrase to “fresh soil subsamples”.

Line 179: Please add a reference for the trait measurements (e.g. Cornelissen et al. 2003 or Harguindeguy et al. 2013).

Lines 303-304: That’s interesting (Ni hyperaccumulation only in FIRE sites).

Lines 306-308: This information is already available on Table 4. The authors could instead focus on the comparison with the TRY database.

Lines 362-364: I can understand the argument, but I would suggest the authors to rewrite this sentence for better clarity.

Lines 365: Replace “anthropized” with “human-dominated”.

Lines 371-377: A comparison with serpentine communities studied in other parts of the world (e.g. Adamidis et al. 2014_PLoS ONE), would be even more interesting/relevant than comparing with the average trait values from the TRY database.

Line 378: Replace “understandable” with “expected”.

Line 383: “Especially”.

Line 388: These results are also congruent with Adamidis et al. 2014_PLoS ONE.

Line 420: “Revegetation” is more specific term than the previously used “reclamation”. Also, another term that could be used instead of reclamation is “restoration”.

Line 420: It would be great if the authors could discuss briefly on this part of the ms, whether these results can be extrapolated and thus used in different disturbed serpentine communities around the world or not.

Line 435: Add a reference supporting the argument of N-fixating species in the community.

Line 445: Replace “from the point of view of” with “in terms of”.

Line 449: Something is wrong here with the reference. Please correct the sentence.

Line 452: Rephrase to “recover nutrients (i.e. nutrient resorption)”.

Lines 453-455: This suggestion is a bit more complicated in my opinion. Introducing N-fixating species might led to colonization from species with more resource-acquisitive traits, that might led to increased competition and maybe to communities with few dominant species. What I mean is that these suggestions are good to be supported with experimental evidence. However, the authors could write this suggestion in a more relaxed manner and support it with references.

Line 461: Maybe the authors could give an example of a mechanism to achieve this.

General comment: It is unexpected that although the two types of disturbance showed highly significant differences in their attributes, the functional diversity between them was similar. I would suggest the authors to discuss a bit more this interesting result. Also, I would suggest to make sure that this result is not the outcome of “the surrounding “source” vegetation” effect mentioned on a previous comment (see comment related to lines 98-103).

Author Response

Dear reviewer #1

Thanks for your comments. Here are our responses. Your comments are in italics, whereas our responses in plain text. Line numbers in our responses correspond to the revised version of the manuscript with visible changes.

The manuscript entitled " Plant Functional Traits on Tropical Ultramafic Habitats Affected by Fire and Mining: Insights for Reclamation" describes a study that investigates differences in soil attributes and species diversity and functional traits in serpentine habitats impacted by different types of disturbance. The study is experimentally well-designed, while the statistical analysis and the replication of the plots is adequate. In general, the manuscript is well-written and easy to follow. Although there is no major weakness in this study, the major comments/suggestions to the authors are related to a better description of the fire history of the burnt sites and of a more detailed description of the surrounding vegetation of the studied sites (serving as the seed source for the colonization of the plots after the disturbance).

Comments/Suggestions

Line 26: Please rephrase to “conservative growth strategy”

Change done (L 26).

Line 35: Please rephrase to “highly plant-available”

Change done (L 35).

Line 42: Please add reference(s)

Reference added (L 42).

Line 49: Rephrase to “Removal of the topsoil limits”, considering that if the capacity was completely removed there would be no vegetation at all after the disturbance.

Modification done. (L 50).

Line 50: I would suggest replacing the adverb “healthy” with a more literal alternative.

Modification done (L 50-51).

Line 53: Please rewrite for better clarity.

Modification done (L 53-55).

Line 54: It could be rephrased as “Tropical forest ecosystems”.

Modification done (L 55).

Line 82: It should be either “Community Weighted Means represent the most probable values” or “Community Weighted Mean represents the most probable value”.

Corrected (L 85).

Line 84: Rephrase to “traits along ecological successions”. Also, replace “on” with “in”.

Following comments from the other reviewer, this sentence has been rephrased to eliminate references to ecological succession (L 87).

Line 85: In lines 81-82 you report that CMW means “Community Weighted Means”. If it doesn’t change, you don’t have to write CWMs to declare the plural.

As we changed definition of CMW to singular, we keep CWMs to declare the plural (L 86).

Line 100: Replace “wildfire” with “wildfires”.

Replaced

Lines 98-103: Did all sites have similar surrounding plant communities. The surrounding vegetation plays a tremendous role as source for the new communities colonizing the disturbed plots. Please provide some more information related to that.

New information included (L 104-113 and L125-128). All plots (except the 4 plots in Bukit Hampuan) are in a disturbed landscape composed by secondary vegetation on different points of succession. In addition to the herbaceous/shrubby vegetation, in the vicinity of all the sampled plots exist sites with more evolved secondary forests. The plots sampled in the Bukit Hampuan Forest Reserve (2 FIRE and 2 MINE), are in the vicinity of primary forests on ultramafic soils. This difference in surrounding vegetation may explain the presence of some species (e.g. the casuarinaceae Ceuthostoma terminale) in those plots. However, species composition and especially levels of taxonomic and functional diversity are similar in BH plots and plots from the other sites.

Lines 100-101: The “frequency of fire” (how many times a site has been burnt in the past) and the “time since the last fire” are two very important parameters affecting the vegetation and thus the functional diversity of plant communities. The authors should provide some more info related to fire history of their plots.

The fire dynamics in Sabah is quite related to the occurrence of the El Niño-Southern Oscillation (ENSO) phenomenon. Two or three times per decade, this phenomenon provokes more severe and longer dry season in Borneo, which in turn favours the spread of humand induced wildfire. The most severe fire events in Sabah occurred on 1983. However, our study area (Bukit Hampuan, Lompoyou and part of Garas) was mainly affected by the fires during the ENSO of 1997/98. These fires affected secondary forests which were previously logged. This info has been included in material and methods (L 104-113 and L125-128). Following a suggestion by the other reviewer, we have fitted environmental information (including time since disturbance) onto the NMDS ordination.

Line 113: Rephrase to “of 10 m radius”.

Corrected (L 123).

Line 128: Rephrase to “fresh soil subsamples”.

Corrected (L 144).

Line 179: Please add a reference for the trait measurements (e.g. Cornelissen et al. 2003 or Harguindeguy et al. 2013).

Reference added (L 199).

Lines 303-304: That’s interesting (Ni hyperaccumulation only in FIRE sites).

Yes, this is interesting. In Sabah, hyperaccumulators tend to occur on secondary forests developed on cambisols (Van der Ent, Chemoecology, 2016). Phyllanthus rufuschaneyi, the hyperaccumulator species we found in some of our plots, is mainly found in secondary forests and in some open areas previously affected by logging and fire. A growth experiment (Nkrumah et al. Plant and Soil 2019) showed that P. rufuschaneyi grown on leptosols (with chemical properties similar to the MINE sites from our study) suffered a reduction in growth compared to plants of the same species grown on cambisols. Future studies should be developed on this issue.

Lines 306-308: This information is already available on Table 4. The authors could instead focus on the comparison with the TRY database.

Lines deleted and a couple of sentences about comparison with TRY database included (L 333-340).

Lines 362-364: I can understand the argument, but I would suggest the authors to rewrite this sentence for better clarity.

Following recommendations by the other reviewer we removed from the manuscript the references to functional diversity and focused on CWMs of the disturbed communities for the determination of interesting traits for remediation.

Lines 365: Replace “anthropized” with “human-dominated”.

This sentence has been rephrased (L 401-403).

Lines 371-377: A comparison with serpentine communities studied in other parts of the world (e.g. Adamidis et al. 2014_PLoS ONE), would be even more interesting/relevant than comparing with the average trait values from the TRY database.

Comparison with Adamidis et al included (L 411-413, L 416-417, L 426-427).

Line 378: Replace “understandable” with “expected”.

Modification done (L 418).

Line 383: “Especially”.

Word removed (L 420).

Line 388: These results are also congruent with Adamidis et al. 2014_PLoS ONE.

Included (L 435).

Line 420: “Revegetation” is more specific term than the previously used “reclamation”. Also, another term that could be used instead of reclamation is “restoration”.

There are different definitions for the widely used terms restoration, reclamation, rehabilitation and remediation. Following the definitions by Lima et al (Environmental Science & Policy, 2016) we avoided the term restoration as we understand that restoration implies the re-establishment of the pre-existing ecosystem, which is seldom feasible in the mid-term. Instead, reclamation aims at the recovery of the ecosystem services provided by the original ecosystem, although the plant communities may be different from the original ecosystem.

Line 420: It would be great if the authors could discuss briefly on this part of the ms, whether these results can be extrapolated and thus used in different disturbed serpentine communities around the world or not.

Comment added (L 481-487).

Line 435: Add a reference supporting the argument of N-fixating species in the community.

References added (L 480).

Line 445: Replace “from the point of view of” with “in terms of”.

Modification done (L 498).

Line 449: Something is wrong here with the reference. Please correct the sentence.

Sorry, it was a typo. Corrected (L 501).

Line 452: Rephrase to “recover nutrients (i.e. nutrient resorption)”.

Modification done (L 504).

Lines 453-455: This suggestion is a bit more complicated in my opinion. Introducing N-fixating species might led to colonization from species with more resource-acquisitive traits, that might led to increased competition and maybe to communities with few dominant species. What I mean is that these suggestions are good to be supported with experimental evidence. However, the authors could write this suggestion in a more relaxed manner and support it with references.

We have modulated our suggestion (L 508-509).

Line 461: Maybe the authors could give an example of a mechanism to achieve this.

One example included (L 515-517).

General comment: It is unexpected that although the two types of disturbance showed highly significant differences in their attributes, the functional diversity between them was similar. I would suggest the authors to discuss a bit more this interesting result. Also, I would suggest to make sure that this result is not the outcome of “the surrounding “source” vegetation” effect mentioned on a previous comment (see comment related to lines 98-103).

Following the suggestion by the other reviewer to reduce the results and discussion section, we have eliminated the references to functional diversity and kept the parts dealing with the CWMs of different traits and its relevance for the trait-assisted selection of suitable plants for reclamation of degraded areas.

Reviewer 2 Report

In this work the authors tried to identify the functional traits 15 of plants that colonise ultramafic areas after disturbance by fire or mining activities. This approach could have important implication for the restoration of degraded areas in ultramafic areas. As a general rule the work is well written, however some parts of the manuscript are a bit long and tedious and not connected each other. For instance, the authors spent lot of time and efforts to describe soil and vegetation differences between MINE and FIRE sites. While such differences are quite obvious (and probably several information could be moved in supplementary materials) they confuse the reader from the main purpose of the work i.e. the identification of the most important functional traits for the two areas. However, despite the strong differences in vegetation and soil patterns between the two sites, the plant traits and the functional diversity differed only slightly. For this reason, I was wondering if the authors chose the wrong traits to investigate.
On the other hand, probably the statistical approach that the authors used is not the most appropriate. I strongly suggest to find a deeper relation between functional traits and vegetation and soil properties of the two kind of sites using multivariate analysis, for instance RDA (or NMDS) analysis. Indeed, in the current version the different parts of the manuscript (soil, vegetation, traits) seem to be disconnected from each other.
Lines 5776 and 84. In the introduction is not clear to me the multiple reference to ecological succession when this work does not treat this aspect.
Line 67-69. I suggest to also cite the use of species traits for restoration purposes and to cite Gilardelli et al. 2015. Ecological Filtering and Plant Traits Variation Across Quarry Geomorphological Surfaces: Implication for Restoration. Environ Manage. 55(5):1147-59.
Line 84. Functional traits, plural
Lines 113-123. How did the authors choose the sampling plot and the direction of the transect? Did the sampling plot have different aspect and slope? Please specify.
Line 193. Did you perform standardization before performing PCA? Please specify
Line198. Do you mean Bray-Curtis distance?
Table 1. Why authors put a note in the table for A, B and C when they can insert directly in the table the three categories?
Table 1. Depth of underground system. I would directly refer to the root system. Please find another expression. In addition, with regard to the investigated traits I was wondering if life forms (according to Rounkier) could give different autcomes. I would expect to find different proportion of growing forms such as annual, perennial, and shroub/wood species; in other word the authors can verify differences in Forbs, Fern, Grasses, Tree, Ginger. I think the author can verify this very quickly.
  Table A1. I was wandering why several species were not identified. This make the work weaker about the possibility to choose the species for restoration.
Results and Discussion are quite long. I think to make the work more readable, it need to be shortened.

Author Response

Dear reviewer #2

Thank you for your comments. Here are our responses. Your comments are marked in italics, whereas our responses in plain text.

Line numbers in our responses correspond to the revised version of the manuscript with visible changes.

In this work the authors tried to identify the functional traits 15 of plants that colonise ultramafic areas after disturbance by fire or mining activities. This approach could have important implication for the restoration of degraded areas in ultramafic areas. As a general rule the work is well written, however some parts of the manuscript are a bit long and tedious and not connected each other. For instance, the authors spent lot of time and efforts to describe soil and vegetation differences between MINE and FIRE sites. While such differences are quite obvious (and probably several information could be moved in supplementary materials) they confuse the reader from the main purpose of the work i.e. the identification of the most important functional traits for the two areas. However, despite the strong differences in vegetation and soil patterns between the two sites, the plant traits and the functional diversity differed only slightly. For this reason, I was wondering if the authors chose the wrong traits to investigate.

As indicated by the reviewer, the aim of the study was to identify the traits of plants that colonise tropical disturbed habitats to enable a future trait-assisted selection of suitable species for reclamation of ultramafic degraded areas. With this aim, we chose a set of functional traits that are interesting for different aspects of reclamation: revegetation and reduction of erosion, nutrient conservation in the ecosystem and possibility of integrating phytomining in the reclamation process. Of course, the slight differences in functional traits between FIRE and MINE plots were unexpected, and in the discussion we proposed different explanations for these results (environmental filter in MINE sites, outcompetition by dominant fern in FIRE sites). We think these results do not invalidate our choice of traits; biological systems are sometimes surprising.

On the other hand, probably the statistical approach that the authors used is not the most appropriate. I strongly suggest to find a deeper relation between functional traits and vegetation and soil properties of the two kind of sites using multivariate analysis, for instance RDA (or NMDS) analysis. Indeed, in the current version the different parts of the manuscript (soil, vegetation, traits) seem to be disconnected from each other.
We disagree with the first sentence, as the statistical methods we applied are widely used in this field of research. Following the suggestion of the reviewer, we have enlarged our analyses to better integrate the three aspects of the manuscript (species composition, functional traits of the community and soil/environemtal aspects) (L238-245):

- first, using the function envfit from package vegan, we have fitted several environmental factors (soil parameters, slope, altitude, aspect and time since disturbance) onto the NMDS ordination based on species abundances.

- second, on the basis of the CWMs of the different functional traits, we performed another NMDS (in this case using Gower distances) and fitted the environmental factors onto this new NMDS ordination using envfit.

In both cases, we plotted on the NMDS only the environmental factors with p-values lower than 0.05

It is interesting to see that in the case of species-based NMDS several factors were correlated to the ordination of plots, whereas in the case of CWM-based NMDS only the soil parameters were correlated to the ordination of plots (L 314-324, L 341-346).

Lines 5776 and 84. In the introduction is not clear to me the multiple reference to ecological succession when this work does not treat this aspect.

Modification done, references to ecological succession eliminated in L 58-59, L 79 and L 87.

Line 67-69. I suggest to also cite the use of species traits for restoration purposes and to cite Gilardelli et al. 2015. Ecological Filtering and Plant Traits Variation Across Quarry Geomorphological Surfaces: Implication for Restoration. Environ Manage. 55(5):1147-59.

Reference added (L 78).

Line 84. Functional traits, plural
Corrected (L 87).

Lines 113-123. How did the authors choose the sampling plot and the direction of the transect? Did the sampling plot have different aspect and slope? Please specify.

For sampling, we chose sites with secondary vegetation developed on serpentinites which had at least 1000 m2 of surface and were accesible. Direction of the first transect from the center of each plot was chosen using a table of random numbers and a compass. Second and third transects were at 120º and 240º angles from first transect. Most of the plots had a main south aspect (from southeast to southwest). There were differences in the slope (FIRE plots are in hill areas with steep slopes, whereas the MINE plots had lower slopes). This information has been included in the revised manuscript. The effect of these environmental factors in the taxonomic and functional composition of plots has been assessed by NMDS with projected environmental vectors (see response to a previous comment above). New information added in L 104-113, L 125-128 and L 130.

Line 193. Did you perform standardization before performing PCA? Please specify
As we already indicated in the original manuscript (L 193-194) we performed the PCA using the correlation matrix between variables due to the different units and magnitudes of each variable. This approach is equivalent to a data standardization, so no additional standardizations are necessary (L 214-215).

Line198. Do you mean Bray-Curtis distance?
Yes, sorry for the mistake. It has been corrected (L 219).

Table 1. Why authors put a note in the table for A, B and C when they can insert directly in the table the three categories?
We did that to fit the table to the page size. Now we have reorganised table 1, inserted the info in the last column and deleted the note (L 205-208).

Table 1. Depth of underground system. I would directly refer to the root system. Please find another expression. In addition, with regard to the investigated traits I was wondering if life forms (according to Rounkier) could give different autcomes. I would expect to find different proportion of growing forms such as annual, perennial, and shroub/wood species; in other word the authors can verify differences in Forbs, Fern, Grasses, Tree, Ginger. I think the author can verify this very quickly.

Depth of underground system has been changed to depth of root system in table 1. We don’t have included the suggestion about life forms in the revised version of the manuscript. We think that the life forms can be redundant with other of the traits we included, such as life cycle (all the sampled species were perennials) or plant height.

Table A1. I was wandering why several species were not identified. This make the work weaker about the possibility to choose the species for restoration.
In some cases plant identification to genus or species was not possible due to lack of reproductive structures and similarity of vegetative organs. However, in most cases this unidentified taxa have a low cover and are present in only one plot, so their contributions to community traits are limited. Moreover, as we indicated in the introduction, the information generated by a ‘functional’ approach can be transferred to other sites with similar conditions, in spite of different local/regional species pool. Thus, our work informs about the traits of plant communities growing on degraded ultramafic areas, and can be transferred to other sites with similar climate.

Results and Discussion are quite long. I think to make the work more readable, it need to be shortened.

We have made efforts to reduce these sections. We have removed the paragraphs and text related to functional diversity of the plots (L 229-236, L 336-338, L 378-400), and kept the parts dealing with CWM data and its relevance for the trait-assisted selection of species for reclamation.

Round 2

Reviewer 2 Report

The author ameliorated the manuscript treating the functional traits of plants that colonise disturbed ultramafic areas subject to Mining or Fire. They accepted several comments of my revision. They did not include some comments but finally, I deem that once an article is sufficiently sound for pubblication, the responsibility of its content is of the authors. I send some other minor comments for typos.

Line 232: Table 2 (upper case)

Line 258: "A_P:Alkaline Phosphatase" (replace colon with comma and insert a space)

Line 279. remove italics from "spp."

Line 282: remove space after "cylindrica"

Line 282: remove italics from sp. and spp. and remove space after "sp."; check spp. and sp. across the manuscript and in Appendix (see Table 1A)

Figure 1: I think that including two small pictures of Mine and Fire areas/soils in Fig. 1 will help the readers to better undestand the two different environments.

Author Response

Dear reviewer,

thank you for your fast feedback. You will find our responses to your comments below. As in the previous round of responses, your comments are in italics, and our responses in plain text.

*********************************************************

The author ameliorated the manuscript treating the functional traits of plants that colonise disturbed ultramafic areas subject to Mining or Fire. They accepted several comments of my revision. They did not include some comments but finally, I deem that once an article is sufficiently sound for pubblication, the responsibility of its content is of the authors. I send some other minor comments for typos.

Line 232: Table 2 (upper case)

Corrected.

Line 258: "A_P:Alkaline Phosphatase" (replace colon with comma and insert a space)

Corrected.

Line 279. remove italics from "spp."

Corrected.

Line 282: remove space after "cylindrica"

Corrected.

Line 282: remove italics from sp. and spp. and remove space after "sp."; check spp. and sp. across the manuscript and in Appendix (see Table 1A)

Corrected and checked in the main text and in Table 1A.

Figure 1: I think that including two small pictures of Mine and Fire areas/soils in Fig. 1 will help the readers to better undestand the two different environments.

Figure 1 has been modified to include two images of FIRE and MINE areas.